# Predicting the aggravation of coronavirus disease-19 pneumonia using chest computed tomography scans

Yukitaka Yamasaki[1], Seido Ooka[2]*, Shin Matsuoka[3], Hayato Tomita[3], Masanori Hirose[4], Tomonori Takano[1], Shotaro Suzuki[2], Mitsuru Imamura[2], Hiroshi Handa[5], Hiroki Nishine[5], Mumon Takita[6], Ayu Minoura[6], Kenichiro Morisawa[6], Takeo Inoue[5], Masamichi Mineshita[5], Kimito Kawahata[2], Hiromu Takemura[7], Shigeki Fujitani[6], Hiroyuki Kunishima[1]

1 Department of Infectious Diseases, St. Marianna University School of Medicine, Miyamae-ku, Kawasaki, Kanagawa, Japan, 2 Division of Rheumatology and Allergology, Department of Internal Medicine, St. Marianna University School of Medicine, Miyamae-ku, Kawasaki, Kanagawa, Japan, 3 Department of Radiology, St. Marianna University School of Medicine, Miyamae-ku, Kawasaki, Kanagawa, Japan, 4 Division of General Internal Medicine, Department of Internal Medicine, St. Marianna University School of Medicine, Miyamae-ku, Kawasaki, Kanagawa, Japan, 5 Division of Respiratory Disease, Department of Internal Medicine, St. Marianna University School of Medicine, Miyamae-ku, Kawasaki, Kanagawa, Japan, 6 Department of Emergency and Critical Care Medicine, St. Marianna University School of Medicine, Miyamae-ku, Kawasaki, Kanagawa, Japan, 7 Department of Microbiology, St. Marianna University School of Medicine, Miyamae-ku, Kawasaki, Kanagawa, Japan

* ooka@marianna-u.ac.jp

**Data Availability Statement:** All relevant data are within the paper and its Supporting Information files.

## Abstract

Presently, coronavirus disease-19 (COVID-19) is spreading worldwide without an effective treatment method. For COVID-19, which is often asymptomatic, it is essential to adopt a method that does not cause aggravation, as well as a method to prevent infection. Whether aggravation can be predicted by analyzing the extent of lung damage on chest computed tomography (CT) scans was examined. The extent of lung damage on pre-intubation chest CT scans of 277 patients with COVID-19 was assessed. It was observed that aggravation occurred when the CT scan showed extensive damage associated with ground-glass opacification and/or consolidation ($p < 0.0001$). The extent of lung damage was similar across the upper, middle, and lower fields. Furthermore, upon comparing the extent of lung damage based on the number of days after onset, a significant difference was found between the severe pneumonia group (SPG) with intubation or those who died and non-severe pneumonia group (NSPG) $\geq$3 days after onset, with aggravation observed when $\geq$14.5% of the lungs exhibited damage at 3–5 days (sensitivity: 88.2%, specificity: 72.4%) and when $\geq$20.1% of the lungs exhibited damage at 6–8 days (sensitivity: 88.2%, specificity: 69.4%). Patients with aggravation suddenly developed hypoxemia after 7 days from the onset; however, chest CT scans obtained in the paucisymptomatic phase without hypoxemia indicated that subsequent aggravation could be predicted based on the degree of lung damage. Furthermore, in subjects aged $\geq$65 years, a significant difference between the SPG and NSPG was observed in the extent of lung damage early beginning from 3 days after onset, and it was found that the degree of lung damage could serve as a predictor of aggravation.

**Funding:** The author(s) received no specific funding for this work.

**Competing interests:** NO authors have competing interests

**Abbreviations:** COVID-19, Coronavirus disease-19; CRP, C-reactive protein; CT, Computed tomography; GGO, Ground-glass opacification; NSPG, Non-severe pneumonia group; ROC, Receiver operating characteristic; SD, Standard deviation; SPG, Severe pneumonia group.

Therefore, to predict and improve prognosis through rapid and appropriate management, evaluating patients with factors indicating poor prognosis using chest CT is essential.

## Introduction

Coronavirus disease-19 (COVID-19) that broke out in 2019 in Wuhan, Hubei Province, China, has spread worldwide [1]. At present, the disease lacks an effective treatment method [2]. As a result, most patients with COVID-19 develop pneumonia, which aggravates in some and becomes life-threatening. Moreover, this troublesome virus often causes an asymptomatic infection and is therefore difficult to prevent [3]. Under such circumstances, preventing aggravation of the disease is most essential.

To this end, it is crucial to accurately understand the state of COVID-19 and to perform early triage of patients whose condition will aggravate. COVID-19 is broadly divided into the asymptomatic phase, in which infection has occurred with no obvious symptoms; the pauci-symptomatic phase, in which flu-like symptoms and symptoms such as taste disorder are exhibited but oxygenation is maintained; and severe pneumonia phases, in which respiratory distress progresses, causing hypoxemia.

If aggravation can be predicted at an early stage before hypoxemia sets in, that is, in the asymptomatic and paucisymptomatic phases, then it would be possible to administer unverified but promising treatments, such as steroid inhalation and antibody cocktail therapy.

Among patients with COVID-19, approximately 3% develop severe pneumonia and die [4]. Previous reports have highlighted that factors that correlate with aggravation include age, underlying illness, and smoking status [5, 6].

However, in reality, only some patients with these factors demonstrate disease aggravation, whereas others do not; thus, these factors are inadequate predictors of aggravation. Additionally, laboratory test values, including lymphocyte count, ferritin level, and C-reactive protein (CRP) level are considered factors that correlate with aggravation [7, 8]. It was previously reported that the lymphocyte count is a predictor of aggravation [9]. However, this predictor is also inadequate as a predictor because it is not a factor directly associated with death caused by pneumonia.

Chest CT scans of patients with pneumonia demonstrate ground-glass opacification (GGO), which is also observed in other viral illnesses. Thus, chest CT scans were also considered as inadequate predictors of aggravation because of the inability to differentiate the causative illnesses. Additionally, in patients with COVID-19, consolidation is observed only in some areas, and the significance of consolidation is unclear.

Therefore, since our institution accepts patients with moderate to severe pneumonia by COVID-19, we investigated what predictors of severe disease were present in patients who consequently transitioned from moderate to severe disease. Particularly, whether chest CT scans can predict the aggravation of pneumonia was examined. Additionally, if aggravation can be predicted, how soon it can be predicted after the appearance of symptoms was examined. Moreover, whether aggravation can be predicted based on the extent of GGO and consolidation was examined qualitatively and quantitatively.

## Materials and methods

### Study subjects

This was a retrospective cohort study. All patients with COVID-19 were diagnosed based on a polymerase chain reaction test on a throat swab. The subject sample included 277

patients with COVID-19 admitted to our hospital from February 16, 2020 to March 9, 2021, and chest CT scans of these patients taken before Ventilator management were evaluated. The clinical course analysis included chest CT scans taken within 14 days of onset. The day of onset was defined as the day of appearance of clinical symptoms [fever or respiratory symptoms (coughing or runny nose) and taste disturbance]. Pneumonia was confirmed based on relevant chest CT scan findings. The severe pneumonia group (SPG) included patients with intubation and conventional ventilator therapy or death. Additionally, patients who were excluded from the SPG were included in the non-severe pneumonia group (NSPG). Patients who were intubated for conditions other than pneumonia were excluded from the evaluations. In many cases, for isolation management, the informed consent was taken orally. This study was conducted with the approval of the institutional review board (approval number 5341).

## The examination of aggravation predictors

The age and gender of the subjects as well as the presence or absence of underlying illnesses in them were evaluated. Blood tests conducted from the day of onset until 14 days thereafter as well as those conducted before intubation were included in evaluations. Tests conducted several times were evaluated using the minimum and maximum values.

Additionally, the white blood cell, lymphocyte, and platelet counts; red cell distribution width; and CRP, ferritin, D-dimer, and sialylated carbohydrate antigen KL-6 levels were evaluated.

The history of use of inhaled steroids including Ciclesonide, non-inhaled steroids, favipiravir, and remdesivir was evaluated as factors affecting the SPG. These evaluations included influential factors found after onset until 48 hours before the day of intubation. The chest CT scans of patients aged ≥65 years were evaluated as age was also considered as one of the factors of aggravation.

## Analysis of chest CT scans

Chest CT scans were independently interpreted and evaluated by two radiologists. The chest CT scans were interpreted separately in terms of the upper, middle, and lower field sections of the left and right sides. The upper field section included slices up to the aortic arch, the middle field section included slices up to the carina, and the lower field section included slices up to 1-cm above the diaphragm. In each of these sections, 1–2 slices were interpreted, the extent of consolidation and GGO was measured, and the mean was calculated. Based on previous reports, the ratios of the upper, middle, and lower lung regions were estimated to be 1:1.6:1.3. [10] Chest CT scans obtained after intubation were excluded. CT severity scoring [11] was calculated per each of the 5 lobes considering the extent of involvement, as follows:

0, no involvement; 1, < 5% involvement; 2, 5–25% involvement; 3, 26–50% involvement; 4, 51–75% involvement; and 5, > 75% involvement. The CT severity score was the sum of each individual lobar score (0 to 25).

## Statistical analysis

Data were analyzed using the Mann–Whitney U test, Fisher's exact test, and regression analysis using Graph Pad Prism 6 (GraphPad Software Inc.).

A receiver operating characteristic (ROC) curve was created using JMP® 15 (SAS Institute Inc., Cary, NC, USA), and the cut-off values for sensitivity and specificity were calculated.

## Results

### Patients

Among the 277 patients, pre-intubation chest CT scans of 65 patients were unavailable; thus, they were excluded from this study. Intubation was performed for myocardial infarction in 3 patients, pulmonary embolism in 4 patients, and gastrointestinal perforation in 1 patient, and thus these patients were excluded from this study.

As a result, 204 patients were included, with 84 patients in the SPG and 120 patients in the NSPG. In the analysis of the clinical course after the onset day, nine patients were excluded due to chest CT scans being taken on day 15 or later after onset, and 6 patients were excluded because the onset day was unknown. Therefore, ultimately, a total of 189 patients were included, of which 76 were included in the SPG and 113 in the NSPG (Fig 1).

### Baseline characteristics

The patient background is indicated in Table 1. The patients had a mean age of 66.1 years, and female patients accounted for 25.8%. Additionally, 78.9% of the patients had an underlying illness, and 37.7% of the patients had diabetes. Intubation was conducted at a mean of 8.1 days after onset (standard deviation (SD) = 4.5). Initial blood sampling was conducted at a mean of 5.7 days (SD = 3.1).

### Examination of aggravation factors

Upon comparing the 84 patients in the SPG and the 120 patients in the NSPG, a significant difference was found in factors that are generally considered to cause aggravation, and the trends in such factors were comparable (Table 2).

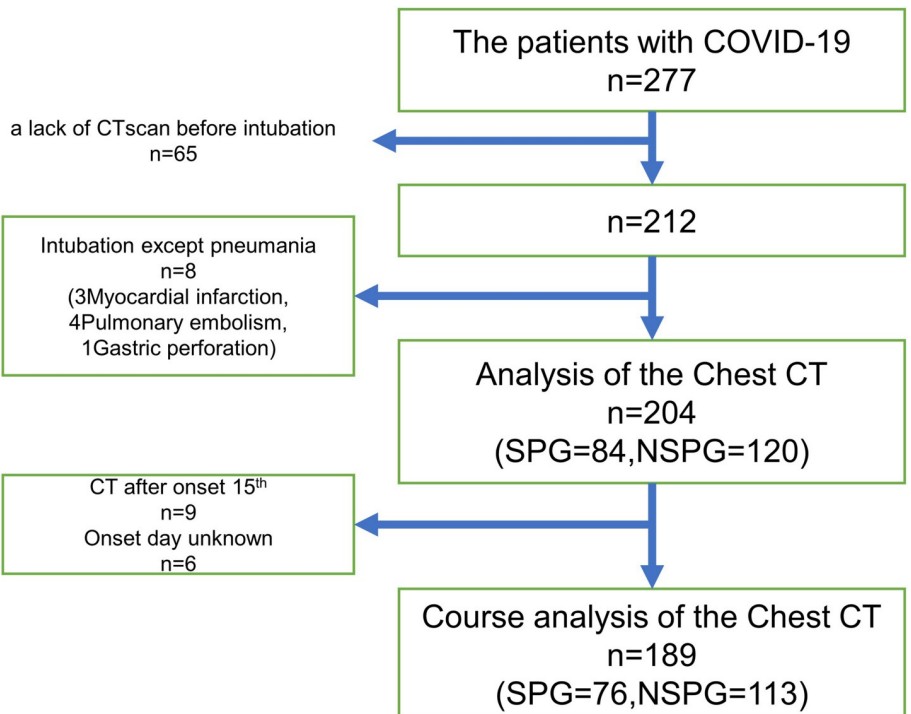

**Fig 1. The study design of this COVID-19 study.** SPG: severe pneumonia group (n = 84); NSPG: non-severe pneumonia group (n = 120).

**Table 1. Baseline characteristics of patients (n = 204).**

| Variable | Value |
|---|---|
| **Age, mean (SD), years** | 66.1 (15.6) |
| **Females, n (%)** | 57 (27.9) |
| **Associated disease, n (%)** | 161 (78.9) |
| **Diabetes, n (%)** | 77 (37.7) |
| **Duration to first blood test, mean (SD), days** | 5.7 (3.1) |
| **Intubation, n (%)** | 71 (34.8) |
| **Duration to intubation, mean (SD), days** | 8.1 (4.5) |

n: number, SD: standard deviation.

Among the aggravation factors for existing reports, there was an insignificant difference between the SPG and NSPG in terms of age and the presence or absence of underlying illnesses.

However, there was a significant difference between the SPG and NSPG ($p = 0.009$) in terms of gender, with more men in the SPG. Consistent with a previous report [9], the lymphocyte count was significantly lower in the SPG ($p = 0.0371$). Furthermore, LDH, CRP, D-dimer, and ferritin levels were significantly higher in the SPG. However, an insignificant difference in terms of KL-6 and hemoglobin A1c was observed between the groups. Patients 65 years or older were more frequent of associated disease than younger patients (90.3% vs 65.2%).

## Chest CT evaluation

On chest CT scans of the SPG and NSPG, a significant difference was observed in the extent of lung damage associated with the findings of GGO combined with consolidation ($p < 0.0001$) (Fig 2A). The ROC curve for the degree of lung damage demonstrated a sensitivity of 72.6%, specificity of 76.0%, and cut-off value of 27.5%. The ROC curve for CT severity score demonstrated a sensitivity of 83.3%, specificity of 62.8%, and cut-off value of 9. Moreover, a significant difference between the SPG and NSPG in terms of the extent of GGO was observed ($p < 0.0001$) (Fig 2B).

**Table 2. Characteristics of patients (n = 204).**

| Variable | SPG (n = 84) | NSPG (n = 120) | P value |
|---|---|---|---|
| **Age, mean (SD), years** | 66.4 (13.93) | 67.5 (16.413) | n.s.**a |
| **Female, n (%)** | 13 (15.3) | 44 (36.7) | 0.009 * |
| **Associated disease, n (%)** | 70 (65.7) | 92 (54.9) | n.s.* |
| **Duration to first blood test, mean (SD), days** | 6.7 (3.6) | 5.2 (3.6) | 0.0027** |
| **Lymphocytes, mean (SD), /mm³** | 729.2 (389.8) | 943.1 (809.5) | 0.0371** |
| **Lactate dehydrogenase, mean (SD), IU/L** | 499.8 (360.0) | 347.1 (148.8) | <0.0001** |
| **C-reactive protein, mean (SD), mg/dL** | 11.6 (8.4) | 6.2 (5.5) | <0.0001** |
| **Sialylated carbohydrate antigen KL-6, mean (SD), U/mL** | 663.9 (576.4) | 459.0 (346.6) | n.s.** |
| **D-dimer, mean (SD), ng/ml** | 16.4 (41.6) | 3.4 (7.2) | 0.0055** |
| **Ferritin, mean (SD), ng/ml** | 1165.3 (1272.6) | 584.5 (488.9) | 0.0009 ** |
| **Hemoglobin A1c, mean (SD), %** | 6.5 (1.4) | 6.8 (1.8) | n.s.** |

*Fisher's exact test;

** Mann–Whitney U test.

n.s.: not significant; n: number, SD: standard deviation.

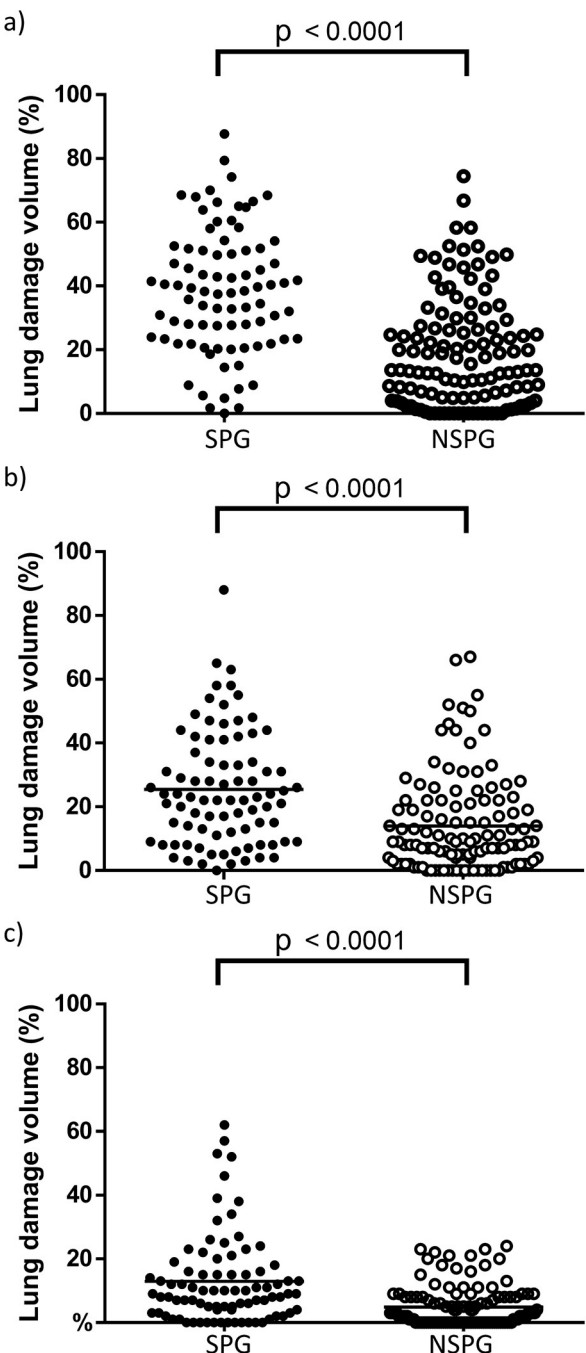

**Fig 2. Comparison of the extent of lung damage on chest CT scans.** a) Overall combination of GGO and consolidation. b) GGO. c) Consolidation.

The ROC curve for GGO revealed a sensitivity of 70.24%, specificity of 65.29%, and a cut-off value of 14.49%.

A significant difference between the groups was also observed in terms of the extent of consolidation ($p < 0.0001$) (Fig 2C). The ROC curve for consolidation showed a sensitivity of 69.05%, specificity of 66.9%, and a cut-off value of 5.13%.

### Analysis according to regions

According to the regions on chest CT, a significant difference in the extent of lung damage between the SPG and NSPG was observed in terms of the area with damage combining GGO and consolidation in the upper, middle, and lower fields (Fig 3A). In the ROC curve of the extent of damage according to regions combining GGO and consolidation, the sensitivity,

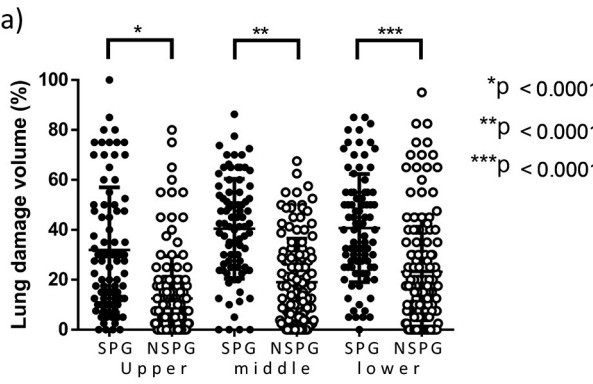

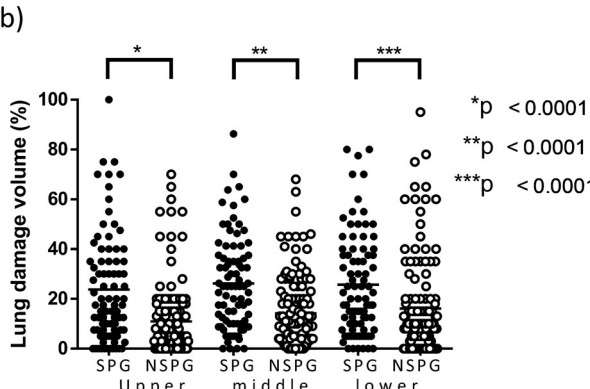

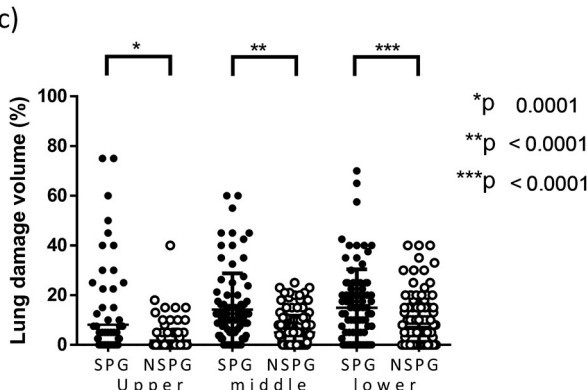

**Fig 3. Comparison of the extent of lung damage according to regions.** a) Overall combining GGO and consolidation. b) GGO. c) Consolidation.

specificity, and cut-off values were 83.3%, 57.8%, and 10.0% for the upper regions; 83.3%, 62.0%, and 22.5%, for the middle field; and 82.14%, 59.5%, and 22.5% for the lower field, respectively.

According to the regions, in all the fields, there was a significant difference in the extent of lung damage between the SPG and NSPG in terms of GGO (Fig 3B). In the ROC curve, the sensitivity, specificity, and cut-off values were 76.2%, 57.9%, and 7.5% for the upper regions; 85.7%, 45.6%, and 7.5% for the middle regions; and 71.4%, 59.5%, and 12.5% for the lower regions, respectively.

Moreover, according to the regions, there was a significant difference in the extent of lung damage between the SPG and NSPG in terms of consolidation (Fig 3C). In the ROC curve, the sensitivity, specificity, and cut-off values were 42.9%, 81.0%, and 2.5% for the upper regions; 63.1%, 76.0%, and 8.8% for the middle regions; and 63.1%, 65.3%, and 7.5% for the lower regions, respectively.

## Clinical course analysis

The time point (X) when a significant difference would appear in findings in CT scans conducted after the day of onset was examined. In the area with damage(Y) combining GGO and consolidation, linear regression was calculated by $Y = 2.822 \times X + 20.18$ and $p = 0.0001$ in the SPG and $Y = 1.353 \times X + 11.93$ and $p = 0.0065$ in the NSPG (Fig 4A). Regarding GGO, linear regression was calculated by $Y = 2.488 \times X + 10.73$ and $p = 0.0055$ in the SPG and $Y = 1.078 \times X + 7.222$ and $p = 0.0054$ in the NSPG. Regarding consolidation, linear regression was not observed in the SPG nor NSPG.

The clinical symptoms of COVID-19 suddenly worsened around day seven of onset, which posed a problem; however, the linear regression analysis revealed a difference between the SPG and NSPG in the early stage on chest CT; therefore, an analysis according to time, dividing the time since onset into four periods (T1 to T4), was conducted. The T1 period was defined as 0–2 days after onset, T2 as 3–5 days after onset, T3 as 6–8 days after onset, and T4 as 9–11 days after onset. The period from T1 to T3 roughly corresponds to the paucisymptomatic phase. In the T1 period, there was an insignificant difference in the extent of lung damage between the SPG and NSPG; however, in the T2 period and after, a significant difference was observed, and it was found that on chest CT taken early after the onset, the area of damage correlates with severe pneumonia (Fig 4B). In the ROC curve, the sensitivity, specificity, and cut-off values were 88.2%, 72.4%, and 14.5% in the T2 period; 78.4%, 69.4%, and 20.1% in the T3 period; and 81.8%, 88.2%, and 33.8% in the T4 period, which indicated that aggravation could be predicted according to the size of the damage area on chest CT. In the ROC curve for CT severity score, the sensitivity, specificity, and cut-off values were 70.6%, 75.0%, and 6 in the T2 period; 76.4%, 67.1%, and 9 in the T3 period; and 90.9%, 83.3%, and 10 in the T4 period.

## Stratification analysis according to aggravation factors

Generally, the prognosis becomes significantly poorer with age; therefore, further analysis of chest CT scans in individuals aged ≥65 years was conducted. Overall, there was a significant difference observed in the chest CT scans in the SPG and NSPG (Fig 5A).

As data under 65 years old, there was not a significant difference observed in the chest CT scans in the SPG and NSPG.

In the clinical course analysis, the point in time when a significant difference would appear in CT findings obtained after the day of onset was examined. As a result, in individuals aged ≥65 in the SPG, linear regression was calculated by $Y = 2.750 \times X + 23.27$ and $p = 0.0109$ (Fig

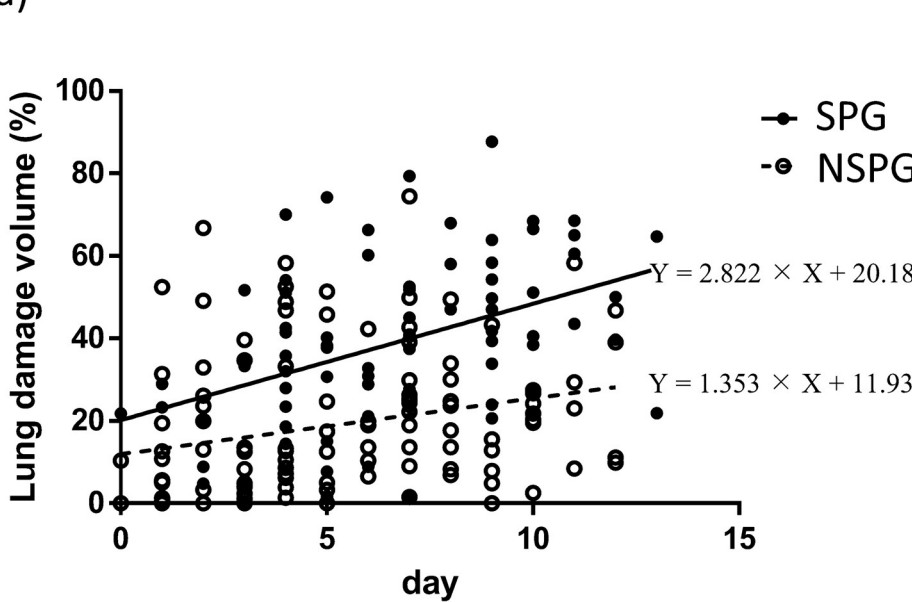

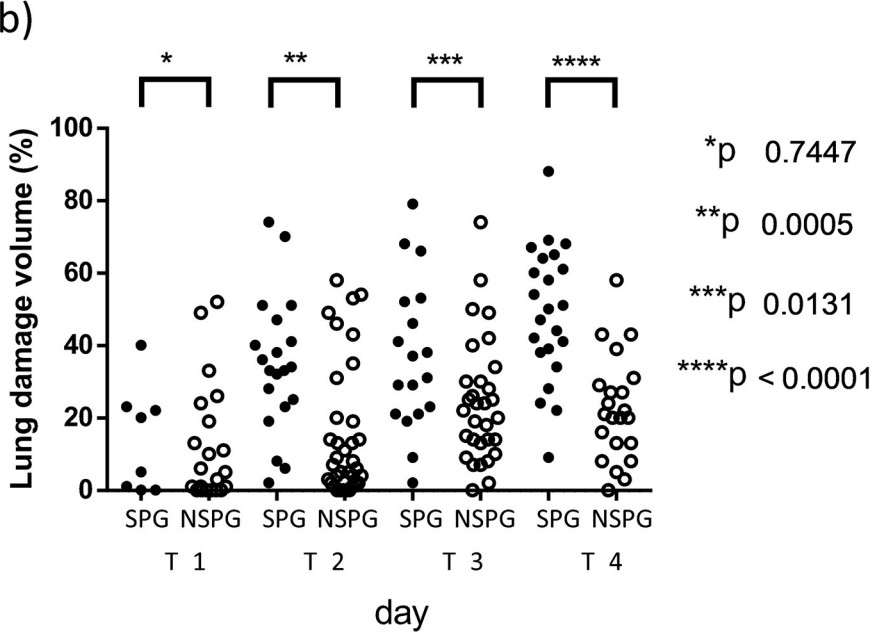

**Fig 4. Comparison of the extent of lung damage according to time after onset.** a) The time from onset and extent of lung damage in the SPG and NSPG. b) Comparison according to time. T1 period: from onset until day 2. T2 period: from day 3 to 5. T3 period: from day 6 to 8. T4 period: from day 9 to 11.

5B) and by $Y = 1.139 \times X + 13.51$ and $p = 0.0504$ in the NSPG. The data under 65 years old did not significantly have linear regression in the SPG and NSPG.

In the analysis according to the various time periods, as per the tendency for all age groups, it was found that on chest CTs taken early after onset from the T2 period onwards, the area of damage correlated with severe pneumonia (Fig 5C).

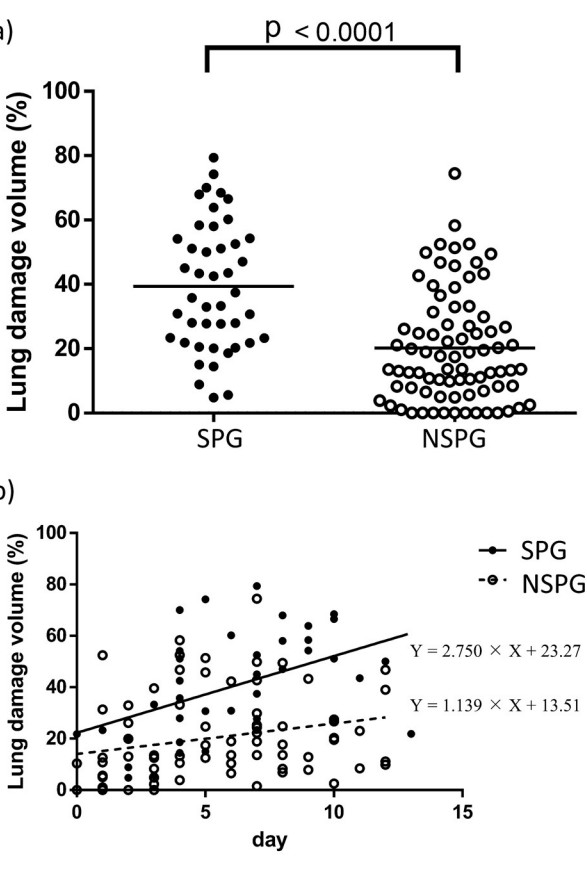

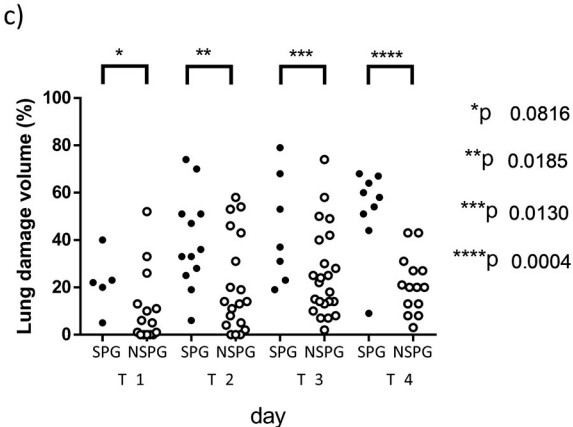

**Fig 5. Comparison of the extent of lung damage in individuals aged ≥65 years.** a) Comparison of the extent of lung damage. b) Time since onset and extent of lung damage in the SPG and NSPG. c) Comparison according to time.

## Discussion

Presently, COVID-19 is spreading worldwide without an effective treatment method. COVID-19 is often asymptomatic; therefore, it is challenging to prevent actual infection [3]. The problem is that some patients with COVID-19 show aggravation of the condition, whereas the condition becomes fatal in some. Therefore, it is believed that individuals with COVID-19 infection should be identified and that it is crucial to adopt an approach that prevents aggravation.

Whether aggravation could be predicted by focusing on the extent of lung damage on CT was examined. It was found that aggravation occurred when there was extensive damage associated with GGO and/or consolidation on chest CT scans with a significant difference as previously reported. Upon comparing the extent of the lung damage according to the regions, comparable results were obtained for the upper, middle, and lower regions. Additionally, on evaluating the extent of lung damage at several days after onset, a significant difference was found between the SPG and NSPG for three days or later after onset, which suggested that the extent of lung damage could serve as a predictor of aggravation. The radiological and clinical progress of pneumonia caused by COVID-19 are consistent [12], and aggravation may be predicted based on the extent of lung damage on early chest CT.

The lymphocyte count is one predictor of aggravation [9]; however, this study examined whether other parameters could also serve as predictors of aggravation. An insignificant difference in age and underlying disease was found between the SPG and NSPG, consistent with previous reports [10, 13]. A previous report showed that advanced age (≥65 years) correlated with the deterioration of the clinical outcomes of patients with COVID-19 [14]; therefore, the extent of lung damage on CT according to age was evaluated. Linear regression was observed on the evaluation of all the age groups, especially in those aged ≥65 years. Additionally, as for all age groups, especially in subjects aged ≥65 years, a significant difference was observed in the extent of lung damage from three days after onset between the SPG and NSPG, and it was found that the extent of lung damage could serve as a predictor of aggravation. Clinically, aggravation occurs from seven days after onset; however, in this study, it was found that the extent of lung damage was increased on chest CT scans obtained earlier. It was believed that the difference in the timing of aggravation found on images obtained before the appearance of clinical symptoms, such as reduced saturation of percutaneous oxygen, is attributed to pulmonary reserve.

Previous research has highlighted that age, underlying illness, and lymphocyte count as well as ferritin, CRP, and D-dimer levels are indicators of aggravation [15, 16], and in this study, apart from an underlying illness, the results were consistent with those of previous reports. It has been reported that GGO correlates with cytokine levels, particularly those of IL-2 [17], which support the spread of lesions in the lungs associated with a cytokine storm in COVID-19. This study is a retrospective study with missing data pertaining to biomarkers. Further, the timing and frequency of the chest CT scans were inconsistent. For higher quality examinations, a prospective study is needed.

It has been highlighted that COVID-19 is often asymptomatic and is difficult to control. Therefore, it is vital to prevent its aggravation. For patients with risk factors for aggravation, it is particularly imperative to evaluate chest CT scans as an indicator of aggravation. Patients whose condition aggravated demonstrated an increase in the extent of lung damage on chest CT scans obtained early after onset. Early CT findings can predict and improve prognosis through rapid and appropriate management.

## Supporting information

**S1 Checklist.**
(DOCX)

**S1 Data.**
(CSV)

## Acknowledgments

We thank all medical staff who treated COVID-19 with us.

The authors have no conflicts of interest directly relevant to the content of this article.

We thank Crimson Interactive Pvt. Ltd. (Ulatus)– www.ulatus.jp for their assistance in manuscript translation and editing.

## Author Contributions

**Conceptualization:** Yukitaka Yamasaki, Seido Ooka, Hiroyuki Kunishima.

**Data curation:** Masanori Hirose, Tomonori Takano.

**Formal analysis:** Yukitaka Yamasaki, Seido Ooka.

**Investigation:** Hiroshi Handa, Hiroki Nishine, Mumon Takita, Ayu Minoura, Kenichiro Morisawa, Takeo Inoue.

**Methodology:** Shin Matsuoka, Hayato Tomita, Shotaro Suzuki, Mitsuru Imamura.

**Validation:** Masamichi Mineshita, Kimito Kawahata, Hiromu Takemura, Shigeki Fujitani, Hiroyuki Kunishima.

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
