## [Decision Letter · Decision Letter 0]

27 Dec 2021

PONE-D-21-37765Predicting the aggravation of coronavirus disease-19 using chest computed tomography scansPLOS ONE

Dear Dr. Seido Ooka,

Thank you for submitting your manuscript to PLOS ONE. After careful consideration, we feel that it has merit but does not fully meet PLOS ONE’s publication criteria as it currently stands. Therefore, we invite you to submit a revised version of the manuscript that addresses the points raised during the review process.

I invited two experts in the field of chest medicine to review this article. The comments are constructive and reasonable. Please revise the MS accordingly or discuss in the proper section if revision was not possible.

We look forward to receiving your revised manuscript.

Kind regards,

Wen-Wei Sung, M.D., Ph.D.

Academic Editor

PLOS ONE

Journal Requirements:

"NO authors have competing interests"

Reviewers' comments:

Reviewer's Responses to Questions

**Comments to the Author**

1. Is the manuscript technically sound, and do the data support the conclusions?

Reviewer #1: Yes

Reviewer #2: Partly

2. Has the statistical analysis been performed appropriately and rigorously? 

Reviewer #1: Yes

Reviewer #2: Yes

3. Have the authors made all data underlying the findings in their manuscript fully available?

Reviewer #1: Yes

Reviewer #2: No

4. Is the manuscript presented in an intelligible fashion and written in standard English?

Reviewer #1: No

Reviewer #2: Yes

5. Review Comments to the Author

Reviewer #1: Summary

Goal of this paper is trying to identify predictors for COVID-19 aggravation in a retrospective clinical analysis. Chest CT scan done by 15 days after disease onset is the key target for this study. 189 candidates were enrolled and divided into SPG(n=76) and NSPG(n=113) groups. It is concluded that the degree of lung damage detected by chest CT scanning in the early stage (3 days after onset) could serve as a predictor for disease aggravation.

Major Issues

(1)

Line 108 : “The severe pneumonia group (SPG) included patients with intubation or respiratory management and those who died.”

----- 1. The definition of “respiratory management” is not precise enough. Does it include oxygen therapy, conventional ventilator therapy, NIPPV, HFNC, prone therapy and/or ECMO? 2. In addition to severe pneumonia with acute respiratory failure, COVID-19 victims may also die of sepsis, cardiac dysfunction, thrombosis or other comorbidities. These cases will be recruited into the SPG group despite their lungs are less damaged, and that will distort the results of chest CT analysis.

(2)

Line 129 : “Analysis of chest CT scans”

-----In this study the methodology of chest CT analysis adopted a scoring system from Muller NL (ref 10) which was specifically for pulmonary sarcoid analysis in 1989. The details of estimation method was not given in this paper. Nevertheless, new expert consensus and chest CT scoring system specifically for COVID -19 infections were available in 2020, which will be more specific and updated for your analysis.

Ref:

# Marco Francone, Franco Iafrate, Giorgio Maria Masci, et al. Chest CT score in COVID-19 patients: correlation with disease severity and short-term prognosis. European Radiology 2020;30(12):6808-6817.

# Feng Pan, Tianhe Ye, Peng Sun, et al. Time Course of Lung Changes at Chest CT during Recovery from Coronavirus Disease 2019 (COVID-19). Radiology 2020; 295:715–721.

# Simpson S, Kay FU, Abbara S, Bhalla S, Chung JH, Chung M, Henry TS, Kanne JP, Kligerman S, Ko JP, Litt H. Radiological Society of North America expert consensus statement on reporting chest CT findings related to COVID-19. Endorsed by the Society of Thoracic Radiology, the American College of Radiology, and RSNA - secondary publication. J Thorac Imaging 2020;35:219-27.

# Geoffrey D. Rubin, Christopher J. Ryerson, Linda B. Haramati, et al. The Role of Chest Imaging in Patient Management during the COVID-19 Pandemic: A Multinational Consensus Statement from the Fleischner Society. Radiology 2020; 296:172–180.

(3)

Line 202-224 : “Analysis according to site”, “…..in terms of the area with damage combining GGO and consolidation in the upper, middle, and lower lobes (Fig 3A).”

-----According to your methodology the so called “the upper, middle, and lower lobes” in this study actually are not the same the terminology of upper, middle, and lower lobes in lung anatomy. Readers will get confused if your “lobe” is actually not a lobe, but a region in the lung.

Minor Issues

Line 225-232 : “Clinical course analysis” “Figure 4A”

----- Simple linear regressions were conducted separately for both SPG and NSPG for different CT lesions. Please specify the Y and X variables in the liner regression equation.

Other Comment

Line 73: “several pneumonia phases”

----- “severe pneumonia” or “several pneumonia” ?

Line 93 :

“As our institution receives patients with moderate to severe pneumonia caused by COVID-19, factors that can predict aggravation from moderate to severe illness in patients who demonstrate the same were examined.”

----- “who demonstrate the same” ???

Line 98 : “was qualitatively examined.”

----- qualitatively or quantitatively or semi-quantitatively examined in this study?

Line 123 : “KL6 levels”

----- KL6 is not a popular abbreviation. Please provide full name in front of abbreviation.

Line 124: “and the use of non-inhaled steroids, including favipiravir and remdesivir”

----- Favipiravir and remdesivir are actually anti-viral agents, not non-inhaled steroids.

Line 290 : “It has been highlighted that COVID-19 is an asymptomatic infection”

----- ???

Reviewer #2: This is a retrospective study. Although 277 COVID-19 patients were accessed initially, 65 patients were excluded because they had no computerized tomography, 8 patients were excluded because they had no pneumonia presented at chest CT scans. Fifteen patients were excluded due to clinical courses were unclear, so only 189 patients were included for course analysis. Given the study designs are not prospective cohort setting, there are some concerns for the interpretation of the results:

Major concern 1: “Age” is an aggravation factors for most of the critical illness, according to existing reports, there was an insignificant difference between the SPG and NSPG in terms of age. The authors should address further analysis of the interaction between age factor and the presence or absence of underlying illnesses.

Figure 5 separately analyzes the individuals who were more than 65 years old, with significant differences. The readers must also request to see this comparison in younger group.

Major concern 2:

There should be other chest computerized tomography image findings, such as interstitial infiltration, septal thickening, cavitating lung lesions, or pneumothorax, etc. The author should try their bet to evaluate possible prognosticator among them.

Major concern 3:

Compared to COVID-19 patients with NSPG, those with SPG overall had higher GGO and consolidation scores; however, their values were overlapping seriously. This makes the results only provide us some insight in realizing pneumonia degree and extension correlate with clinical severity. It will be more practical to develop a chest computerized tomography scoring system to predict clinal outcomes. Please refer to the published paper “Chest CT Severity Score: An Imaging Tool for Assessing Severe COVID-19” at Radiology: Cardiothoracic Imaging Volume 2: Number 2, 2020.

Minor concern 1:

Since the 65 patients without CT scans and the 8 patients without pneumonia presented at chest CT scans were excluded, the title should be verified as “The aggravation of coronavirus disease-19 pneumonia in chest computed tomography scans”.

Minor concern 2:

Figure 2 to Figure 5 should annotate the longitudinal axis meaning to indicate what kind of percentage. Figure 2A lacks horizontal bar to indicate mean values.

6. PLOS authors have the option to publish the peer review history of their article (what does this mean?). If published, this will include your full peer review and any attached files.

Reviewer #1: No

Reviewer #2: No

---

## [Author Response · Author response to Decision Letter 0]

8 Apr 2022

Response to Reviewers' comments:

Reviewer #1: Summary

Goal of this paper is trying to identify predictors for COVID-19 aggravation in a retrospective clinical analysis. Chest CT scan done by 15 days after disease onset is the key target for this study. 189 candidates were enrolled and divided into SPG(n=76) and NSPG(n=113) groups. It is concluded that the degree of lung damage detected by chest CT scanning in the early stage (3 days after onset) could serve as a predictor for disease aggravation.

⇒We thank the reviewer for the advice. 

Major Issues

(1)

Line 108 : “The severe pneumonia group (SPG) included patients with intubation or respiratory management and those who died.”

----- 

1. The definition of “respiratory management” is not precise enough. Does it include oxygen therapy, conventional ventilator therapy, NIPPV, HFNC, prone therapy and/or ECMO? 

⇒The definition of "respiratory management" is ventilator management.

Therefore, the definition of "severe pneumonia" in this study is ventilator management or death.

It is described in lines 143-144.

The severe pneumonia group (SPG) included patients with intubation and conventional ventilator therapy or death.

2. In addition to severe pneumonia with acute respiratory failure, COVID-19 victims may also die of sepsis, cardiac dysfunction, thrombosis or other comorbidities. These cases will be recruited into the SPG group despite their lungs are less damaged, and that will distort the results of chest CT analysis.

⇒ "Myocardial infarction" and "pulmonary embolism" are excluded in this study.

It is described in lines 145-146.

Patients who were intubated for conditions other than pneumonia were excluded from the evaluations.

It is described in lines 186-188.

Intubation was performed for myocardial infarction in 3 patients, pulmonary embolism in 4 patients, and gastrointestinal perforation in 1 patient, and thus these patients were excluded from this study.

(2)

Line 129 : “Analysis of chest CT scans”

-----In this study the methodology of chest CT analysis adopted a scoring system from Muller NL (ref 10) which was specifically for pulmonary sarcoid analysis in 1989. The details of estimation method was not given in this paper. Nevertheless, new expert consensus and chest CT scoring system specifically for COVID -19 infections were available in 2020, which will be more specific and updated for your analysis.

Ref:

# Marco Francone, Franco Iafrate, Giorgio Maria Masci, et al. Chest CT score in COVID-19 patients: correlation with disease severity and short-term prognosis. European Radiology 2020;30(12):6808-6817.

# Feng Pan, Tianhe Ye, Peng Sun, et al. Time Course of Lung Changes at Chest CT during Recovery from Coronavirus Disease 2019 (COVID-19). Radiology 2020; 295:715–721.

# Simpson S, Kay FU, Abbara S, Bhalla S, Chung JH, Chung M, Henry TS, Kanne JP, Kligerman S, Ko JP, Litt H. Radiological Society of North America expert consensus statement on reporting chest CT findings related to COVID-19. Endorsed by the Society of Thoracic Radiology, the American College of Radiology, and RSNA - secondary publication. J Thorac Imaging 2020;35:219-27.

# Geoffrey D. Rubin, Christopher J. Ryerson, Linda B. Haramati, et al. The Role of Chest Imaging in Patient Management during the COVID-19 Pandemic: A Multinational Consensus Statement from the Fleischner Society. Radiology 2020; 296:172–180.

⇒The CT severity scoring of 2020 is had scoring of for each lobe. Therefore the right middle lobe of approximately 10% of total lung is actually estimated as 20% by CT severity scoring. As CT severity scoring was not exact for an area, we chose a standard method.

However, in consideration of the need of the on-site physician who used CT severity scoring, we wrote score jointly.

It is described in lines 171-175.

 CT severity scoring [11] was calculated per each of the 5 lobes considering the extent of involvement, as follows:

0, no involvement; 1, < 5% involvement; 2, 5-25% involvement; 3, 26-50% involvement; 4, 51-75% involvement; and 5, > 75% involvement. The CT severity score was the sum of each individual lobar score (0 to 25).

It is described in lines 226-227.

The ROC curve for CT severity score demonstrated a sensitivity of 83.3%, specificity of 62.8%, and cut-off value of 9.

It is described in lines 287-289.

In the ROC curve for CT severity score, the sensitivity, specificity, and cut-off values were 70.6%, 75.0%, and 6 in the T2 period; 76.4%, 67.1%, and 9 in the T3 period; and 90.9%, 83.3%, and 10 in the T4 period.

(3)

Line 202-224 : “Analysis according to site”, “…..in terms of the area with damage combining GGO and consolidation in the upper, middle, and lower lobes (Fig 3A).”

-----According to your methodology the so called “the upper, middle, and lower lobes” in this study actually are not the same the terminology of upper, middle, and lower lobes in lung anatomy. Readers will get confused if your “lobe” is actually not a lobe, but a region in the lung.

⇒The expression "lobe" and “site” has been changed to "regions".

Minor Issues

Line 225-232 : “Clinical course analysis” “Figure 4A”

----- Simple linear regressions were conducted separately for both SPG and NSPG for different CT lesions. Please specif the y Y and X variables in the liner regression equation.

⇒In Figure 4a, we added Y = 2.822 × X + 20.18 as SPG and Y = 1.353 × X + 11.93 as NSPG.

In Figure 5b, we added Y = 2.750 × X + 23.27 as SPG and Y = 1.139 × X + 13.51 as NSPG

Other Comment

Line 73: “several pneumonia phases”

----- “severe pneumonia” or “several pneumonia” ?

⇒The expression " several pneumonia" has been changed to " severe pneumonia” in line 107

Line 93 :

“As our institution receives patients with moderate to severe pneumonia caused by COVID-19, factors that can predict aggravation from moderate to severe illness in patients who demonstrate the same were examined.”

----- “who demonstrate the same” ???

⇒In 126, we added 

“Therefore, since our institution accepts patients with moderate to severe pneumonia by COVID-19, we investigated what predictors of severe disease were present in patients who consequently transitioned from moderate to severe disease.”

Line 98 : “was qualitatively examined.”

----- qualitatively or quantitatively or semi-quantitatively examined in this study?

In line132, we changed as “was examined qualitatively and quantitatively.”migi 

Line 123 : “KL6 levels”

----- KL6 is not a popular abbreviation. Please provide full name in front of abbreviation.

⇒In line156, we changed as “sialylated carbohydrate antigen KL-6”

Line 124: “and the use of non-inhaled steroids, including favipiravir and remdesivir”

----- Favipiravir and remdesivir are actually anti-viral agents, not non-inhaled steroids.

Inlind157, we added “The history of use of inhaled steroids including Ciclesonide, non-inhaled steroids, favipiravir, and remdesivir was evaluated as factors affecting the severe pneumonia group.”

Line 290 310: “It has been highlighted that COVID-19 is an asymptomatic infection”

----- 

⇒In Line 349, we changed to “It has been highlighted that COVID-19　is often asymptomatic and is difficult to control.”

Reviewer #2: This is a retrospective study. Although 277 COVID-19 patients were accessed initially, 65 patients were excluded because they had no computerized tomography, 8 patients were excluded because they had no pneumonia presented at chest CT scans. Fifteen patients were excluded due to clinical courses were unclear, so only 189 patients were included for course analysis. Given the study designs are not prospective cohort setting, there are some concerns for the interpretation of the results:

⇒We thank the reviewer for the advice. 

Major concern 1: “Age” is an aggravation factors for most of the critical illness, according to existing reports, there was an insignificant difference between the SPG and NSPG in terms of age. The authors should address further analysis of the interaction between age factor and the presence or absence of underlying illnesses.

Figure 5 separately analyzes the individuals who were more than 65 years old, with significant differences. The readers must also request to see this comparison in younger group.

⇒In line 220, we added “Patients 65 years or older were more frequent of associated disease than younger patients (90.3% vs 65.2%).”

In line 299, we added “As data under 65 years old, there was not a significant difference observed in the chest CT scans in the SPG and NSPG,”

In line 304, we added “The data under 65 years old did not significantly have linear regression in the SPG and NSPG.,”

Major concern 2:

There should be other chest computerized tomography image findings, such as interstitial infiltration, septal thickening, cavitating lung lesions, or pneumothorax, etc. The author should try their bet to evaluate possible prognosticator among them.

⇒ These findings were rare and did not make a significant difference. Therefore the extent of consolidation and GGO was measured, and the mean was calculated.

Major concern 3:

Compared to COVID-19 patients with NSPG, those with SPG overall had higher GGO and consolidation scores; however, their values were overlapping seriously. This makes the results only provide us some insight in realizing pneumonia degree and extension correlate with clinical severity. It will be more practical to develop a chest computerized tomography scoring system to predict clinal outcomes. Please refer to the published paper “Chest CT Severity Score: An Imaging Tool for Assessing Severe COVID-19” at Radiology: Cardiothoracic Imaging Volume 2: Number 2, 2020.

⇒The CT severity scoring of 2020 is had scoring of for each lobe. Therefore the right middle lobe of approximately 10% of total lung is actually estimated as 20% by CT severity scoring. As CT severity scoring was not exact for an area, we chose a standard method.

However, in consideration of the need of the on-site physician who used CT severity scoring, we wrote score jointly.

It is described in lines 171-175.

 CT severity scoring [11] was calculated per each of the 5 lobes considering the extent of involvement, as follows:

0, no involvement; 1, < 5% involvement; 2, 5-25% involvement; 3, 26-50% involvement; 4, 51-75% involvement; and 5, > 75% involvement. The CT severity score was the sum of each individual lobar score (0 to 25).

It is described in lines 226-227.

The ROC curve for CT severity score demonstrated a sensitivity of 83.3%, specificity of 62.8%, and cut-off value of 9.

It is described in lines 287-289.

In the ROC curve for CT severity score, the sensitivity, specificity, and cut-off values were 70.6%, 75.0%, and 6 in the T2 period; 76.4%, 67.1%, and 9 in the T3 period; and 90.9%, 83.3%, and 10 in the T4 period.

Minor concern 1:

Since the 65 patients without CT scans and the 8 patients without pneumonia presented at chest CT scans were excluded, the title should be verified as “The aggravation of coronavirus disease-19 pneumonia in chest computed tomography scans”.

⇒We changed the title to “The aggravation of coronavirus disease-19 pneumonia in chest computed tomography scans”

Minor concern 2:

Figure 2 to Figure 5 should annotate the longitudinal axis meaning to indicate what kind of percentage. Figure 2A lacks horizontal bar to indicate mean values.

⇒ We added “Lung damage volume (%)” in Figure 2 to Figure 5

---

## [Decision Letter · Decision Letter 1]

2 May 2022

PONE-D-21-37765R1Predicting the aggravation of coronavirus disease-19 pneumonia using chest computed tomography scansPLOS ONE

Dear Dr. Seido Ooka,

Thank you for submitting your manuscript to PLOS ONE. After careful consideration, we feel that it has merit but does not fully meet PLOS ONE’s publication criteria as it currently stands. Therefore, we invite you to submit a revised version of the manuscript that addresses the points raised during the review process.

We look forward to receiving your revised manuscript.

Kind regards,

Wen-Wei Sung, M.D., Ph.D.

Academic Editor

PLOS ONE

Journal Requirements:

Reviewers' comments:

Reviewer's Responses to Questions

**Comments to the Author**

1. If the authors have adequately addressed your comments raised in a previous round of review and you feel that this manuscript is now acceptable for publication, you may indicate that here to bypass the “Comments to the Author” section, enter your conflict of interest statement in the “Confidential to Editor” section, and submit your "Accept" recommendation.

Reviewer #1: All comments have been addressed

2. Is the manuscript technically sound, and do the data support the conclusions?

Reviewer #1: Yes

3. Has the statistical analysis been performed appropriately and rigorously? 

Reviewer #1: Yes

4. Have the authors made all data underlying the findings in their manuscript fully available?

Reviewer #1: Yes

5. Is the manuscript presented in an intelligible fashion and written in standard English?

Reviewer #1: Yes

6. Review Comments to the Author

Reviewer #1: Comments

Minor Issues:

Line 205 Table 1

The superscript characters of “a” and “b” represent “number” and “standard deviation” respectively in Table 1. However, there are already abbreviations of “n” and “SD” to represent “number” and “standard deviation” in the same table. The superscript characters of “a” and “b” are superfluous.

Line 215 Table 2

The superscript characters of “a”, “b” and “c” represent “not significant” and “number” and “standard deviation” respectively in Table 2. However, there are already abbreviations of “n.s.”, “n” and “SD” in the table. The superscript characters of “a”, “b” and “c” are superfluous.

Line 251

“middlefield” →“middle field”

Line 268

“The time point when a significant difference would appear in findings in CT scans conducted after the day of onset was examined.” →“The time point (X) when a significant difference would appear in findings in CT scans conducted after the day of onset was examined. ”

Line 269

“ In the area with damage combining GGO and consolidation,” →“In the area with damage(Y) combining GGO and consolidation,”

7. PLOS authors have the option to publish the peer review history of their article (what does this mean?). If published, this will include your full peer review and any attached files.

Reviewer #1: No

---

## [Author Response · Author response to Decision Letter 1]

28 Sep 2022

Thank you very much for your reviewing. We totally agree to your suggestions and re-vised the text. Revised portions were colored red. We believe that this revision made the text better than before. Please, consider my article for publication in your journal.

Reviewers' comments:

Reviewer #1:

Line 205 Table 1

The superscript characters of “a” and “b” represent “number” and “standard deviation” respectively in Table 1. However, there are already abbreviations of “n” and “SD” to represent “number” and “standard deviation” in the same table. The superscript charac-ters of “a” and “b” are superfluous.

⇒We thank the reviewer for the advice. 

I deleted the superscript characters of "a" and "b"

Line 215 Table 2

The superscript characters of “a”, “b” and “c” represent “not significant” and “number” and “standard deviation” respectively in Table 2. However, there are already abbrevia-tions of “n.s.”, “n” and “SD” in the table. The superscript characters of “a”, “b” and “c” are superfluous.

⇒We thank the reviewer for the advice. 

I deleted the superscript characters of "a" and "b"

Line 251

“middlefield” →“middle field”

⇒We thank the reviewer for the advice. 

I corrected it according to your advice.

Line 268

“The time point when a significant difference would appear in findings in CT scans con-ducted after the day of onset was examined.” →“The time point (X) when a significant difference would appear in findings in CT scans conducted after the day of onset was examined. ”

⇒We thank the reviewer for the advice. 

I corrected it according to your advice.

Line 269

“ In the area with damage combining GGO and consolidation,” →“In the area with dam-age(Y) combining GGO and consolidation,”

⇒We thank the reviewer for the advice. 

I corrected it according to your advice.

---

## [Editor Report · Decision Letter 2]

13 Oct 2022

Predicting the aggravation of coronavirus disease-19 pneumonia using chest computed tomography scans

PONE-D-21-37765R2

Dear Dr. Seido Ooka,

We’re pleased to inform you that your manuscript has been judged scientifically suitable for publication and will be formally accepted for publication once it meets all outstanding technical requirements.

Kind regards,

Wen-Wei Sung, M.D., Ph.D.

Academic Editor

PLOS ONE

---

## [Editor Report · Acceptance letter]

26 Oct 2022

PONE-D-21-37765R2 

Predicting the aggravation of coronavirus disease-19 pneumonia using chest computed tomography scans 

Dear Dr. Ooka:

I'm pleased to inform you that your manuscript has been deemed suitable for publication in PLOS ONE. Congratulations! Your manuscript is now with our production department. 

Kind regards, 

on behalf of

Dr. Wen-Wei Sung 

Academic Editor

PLOS ONE